# Exploring the Promoter Generation and Prediction of *Halomonas* spp. Based on GAN and Multi-Model Fusion Methods

**DOI:** 10.3390/ijms252313137

**Published:** 2024-12-06

**Authors:** Cuihuan Zhao, Yuying Guan, Shuan Yan, Jiahang Li

**Affiliations:** 1Center for Synthetic and Systems Biology, School of Life Sciences, Tsinghua University, Beijing 100084, China; zhaocuihuan@mail.tsinghua.edu.cn (C.Z.);; 2Department of Engineering Physics, Institute of Public Safety Research, Tsinghua University, Beijing 100084, China; 3School of Mathematical Sciences, Nankai University, Tianjin 300071, China

**Keywords:** *Halomonas*, promoters, generative adversarial networks (GANs), multi-model fusion, quantile hit rate

## Abstract

Promoters, as core elements in the regulation of gene expression, play a pivotal role in genetic engineering and synthetic biology. The accurate prediction and optimization of promoter strength are essential for advancing these fields. Here, we present the first promoter strength database tailored to *Halomonas*, an extremophilic microorganism, and propose a novel promoter design and prediction method based on generative adversarial networks (GANs) and multi-model fusion. The GAN model effectively learns the key features of *Halomonas* promoter sequences, such as the GC content and Moran’s coefficients, to generate biologically plausible promoter sequences. To enhance prediction accuracy, we developed a multi-model fusion framework integrating deep learning and machine learning approaches. Deep learning models, incorporating BiLSTM and CNN architectures, capture k-mer and PSSM features, whereas machine learning models utilize engineered string and non-string features to construct comprehensive feature matrices for the multidimensional analysis and prediction of promoter strength. Using the proposed framework, newly generated promoters via mutation were predicted, and their functional validity was experimentally confirmed. The integration of multiple models significantly reduced the experimental validation space through an intersection-based strategy, achieving a notable improvement in top quantile prediction accuracy, particularly within the top five quantiles. The robustness and applicability of this model were further validated on diverse datasets, including test sets and out-of-sample promoters. This study not only introduces an innovative approach for promoter design and prediction in *Halomonas* but also lays a foundation for advancing industrial biotechnology. Additionally, the proposed strategy of GAN-based generation coupled with multi-model prediction demonstrates versatility, offering a valuable reference for promoter design and strength prediction in other extremophiles. Our findings highlight the promising synergy between artificial intelligence and synthetic biology, underscoring their profound academic and practical implications.

## 1. Introduction

In the realms of gene regulation and synthetic biology, promoters serve as critical regulatory elements that dictate gene expression levels. Their strength and characteristics directly influence the expression efficiency of target genes and the yields of downstream metabolites [1,2,3]. However, the inherent complexity of promoter sequences and the significant biological variability across species make the rapid and efficient design and selection of promoters with desired strengths a longstanding challenge in synthetic biology [4,5]. This challenge is particularly pronounced in extremophilic microorganisms such as *Halomonas*, where the unique genomic architectures and regulatory networks have left promoter studies and applications in a nascent stage [2,6].

*Halomonas* is a class of Gram-negative, salinophilic, alkaliphilic, rod-shaped bacteria that can survive at a salt concentration of 60 g/L and a pH of 8–9. It as a non-modal extremophile microorganism that is widely used in the fields of biomaterials (e.g., PHA), biosurfactants, and other fields. In addition, its salt tolerance gives it a significant competitive advantage in open fermentation systems, reducing production costs. What is special about the promoters of *Halomonas* is that they must be adapted to the high-salt and alkaline environment in which the host cells are found. Promoters are DNA sequences upstream of genes that are responsible for regulating the initiation of gene expression. In *Halomonas*, the promoter core regions (−10 and −35 regions) typically retain sequence features similar to those of other Gram-negative bacteria, but because they are required to ensure normal gene expression under high-salt and high-alkaline conditions, *Halomonas* promoters may contain unique gene structures to be able to drive higher levels of gene expression under high osmolarity conditions. Therefore, studying *Halomonas* promoters and their stability in complex environments could optimize the expression of key enzymes, enhance the yield of targeted metabolites, and provide a broad scope for the development of novel bioproducts and green industrial production processes.

Recent advancements in high-throughput sequencing and deep learning methodologies have led to significant progress in promoter strength prediction. Traditional promoter design and evaluation approaches have relied heavily on experimental validation, such as molecular biology techniques that modify and screen promoters by introducing random variations through chemical treatments [7,8] or insertional mutagenesis [9]. These methods have enabled the creation of diverse promoter libraries [10], which have subsequently been assessed using reporter genes, such as fluorescent proteins or enzymatic activity genes, to measure transcriptional activity. Although these foundational methods have provided reliable functional validation, they are inherently labor-intensive, time-consuming, costly, and prone to unpredictability, limiting their scalability and broader applicability.

The integration of machine learning and deep learning in recent years has introduced transformative improvements in promoter strength prediction. By employing feature extraction and sequence embedding techniques, traditional machine learning models such as support vector machines (SVMs) [11] and random forests (RFs) [11] have been augmented by advanced deep learning frameworks like convolutional neural networks (CNNs) [12] and long short-term memory networks (LSTMs) [13]. These models autonomously extract features from promoter sequences, enabling precise strength predictions. However, existing efforts predominantly focus on model organisms like *Escherichia coli* and yeast, leaving significant gaps in research on non-model microorganisms such as *Halomonas*. Moreover, the limitations of current databases and feature extraction methodologies constrain the scope and depth of *Halomonas* promoter studies. Additionally, current predictive methods often target binary classification tasks (e.g., distinguishing promoters from non-promoters or categorizing promoter strength into strong or weak classes) [11,12,13,14,15,16,17,18,19,20,21,22,23,24,25,26], which fail to capture the continuous relationship between promoter sequences and strength. To address this gap, more accurate regression models are needed to establish these quantitative relationships.

This study proposes an integrative framework combining generative adversarial networks (GANs), deep learning models (BiLSTM and CNN), and machine learning algorithms (random forest) to address the design and strength prediction of *Halomonas* promoters. The framework provides a theoretical foundation and technical toolkit for *Halomonas* and other extremophilic promoter engineering. The key contributions include (1) the construction and optimization of a *Halomonas* promoter database to extract characteristic distributions and establish biologically meaningful representations; (2) the development of a GAN-based promoter generation network to design biologically relevant candidate promoters; (3) the creation of a hybrid prediction model by integrating deep learning (BiLSTM and CNN) with traditional machine learning (random forest), enabling accurate candidate promoter selection; and (4) the experimental validation of candidate promoters’ functional strengths to assess the practical applicability of the predictive model.

The overall framework, depicted in Figure 1, comprises three key components: promoter design, strength prediction, and validation. Initially, a promoter strength database for *Halomonas* species was constructed. In the promoter generation phase, promoters were synthesized using a generative adversarial network (GAN), wherein the GAN generator was responsible for producing noisy promoter sequences and the discriminator evaluated whether these promoters met the target specifications by calculating the loss between the noisy and real promoters. Subsequently, the promoters generated by the discriminator underwent an assessment of their biological characteristics, including Moran’s coefficient and the GC content. For the promoter prediction segment, a fusion predictive model integrating two distinct categories of models was employed. Firstly, deep learning models were trained on promoter datasets from both *Halomonas* and *E. coli* using k-mer and position-specific scoring matrix (PSSM) features, respectively. These deep learning models comprised two separate networks: bidirectional long short-term memory (BiLSTM) and convolutional neural networks (CNNs). Concurrently, machine learning models, specifically random forests, were trained on the *Halomonas* promoter dataset utilizing both string-based and non-string-based feature engineering techniques. The fusion of these two modeling approaches constituted the proposed promoter strength prediction framework. Finally, the promoters generated by the GAN were subjected to rigorous screening, and the strengths of 95 novel mutant promoters were verified by experimental data. Comparative analyses revealed that the fusion model significantly enhanced the quantile hit rates in promoter strength predictions, underscoring its efficacy in accurately forecasting promoter activity.

This study presents a novel approach to rapid, efficient, and accurate promoter screening, providing theoretical support for *Halomonas* promoter design and offering valuable insights for optimizing promoters in other microorganisms. The findings hold substantial theoretical significance and practical utility, advancing the frontiers of synthetic biology and biotechnology.

## 2. Results

### 2.1. Development and Validation of the Promoter Prediction Model

The proposed framework consists of two major components: promoter design and promoter strength prediction. To design functional promoters, we employed a DNA sequence generator based on a generative adversarial network (GAN). The GAN model, comprising a generator and a discriminator, was trained to simulate DNA sequence generation. Through an iterative feedback mechanism, the generated DNA sequences were optimized to match specific target characteristics, such as the GC content and Moran’s coefficient. Sequences meeting the predefined criteria were selected, allowing for the generation of promoters with specified properties and in desired quantities or categories.

For promoter strength prediction, two distinct predictive models were developed. The first is a deep learning-based prediction model. This deep learning model comprises two independent input streams built using promoter datasets from *Halomonas* bacteria and *E. coli*. All promoters were standardized to a length of 50 bp, and features were extracted using two approaches: Word2Vec-based k-mer features and CGR-based PSSM matrix features. These feature sets were input into the deep learning model, resulting in the following performance: Spearman correlation: 0.5266, *p*-value: 2.77 × 10^−109^; Pearson correlation: 0.5699, *p*-value: 9.69 × 10^−132^.

The second model is a RandomForestRegressor-based prediction model. This machine learning model utilized only the *Halomonas* bacterial promoter dataset and its corresponding strengths, excluding the *E. coli* dataset. Input features were constructed through string-based and non-string-based feature engineering methods. String-based features: “CountVectorizer”, “CountVectorizer_n_grams”, “TfidfVectorizer”, “HashingVectorizer”, “LDA”, “LSA”, “PCA”, “t-SNE”, “Word2Vec”, “FastText”, “Doc2Vec”, “GloVe”, and “BERT”. Non-string-based features: the GC content, Moran’s coefficient, CKSNAP features, EIIP features, and k-mers (k = 2, 3, 4, 5, 6). These features were combined into a single feature matrix for input into the RandomForestRegressor model, achieving the following performance: Spearman correlation: 0.0984, *p*-value: 0.0730; Pearson correlation: 0.1240, *p*-value: 0.0236.

The feature construction methodologies for the machine learning model and the deep learning model differ substantially, leading to distinct predictive outcomes. Notably, the performance of the RandomForestRegressor machine learning model is inferior to that of the deep learning model. The deep learning model leverages high-dimensional features derived from graph-based information and graph convolution, whereas the RandomForestRegressor model utilizes features constructed through string-based and non-string-based engineering techniques. These features are distinct from those employed in the deep learning model, representing a novel feature construction approach. Each feature generated for the RandomForestRegressor corresponds to a one-dimensional sequence result, and the integration of all features is required to form a comprehensive feature input matrix. Consequently, the methods for feature computation and the resulting feature representations are entirely different from those of the deep learning model. It is important to note that the high-dimensional features employed in deep learning refer to those with more than two dimensions. Specifically, the k-mer features utilized are three-dimensional, whereas the position-specific scoring matrix (PSSM) features are four-dimensional. In contrast, the features used in machine learning consist of two-dimensional matrices.

Additionally, the deep learning model incorporates the *E. coli* dataset, whereas the RandomForestRegressor model does not. This differentiation allows each model to perceive data characteristics uniquely, promoting diversity in feature utilization. Such differentiation facilitates the integration of results from multiple models, enabling cross-validation and enhancing the screening of high-performance promoters.

The promoter generation network is capable of producing millions of promoters with ease. However, the number of promoters that can be experimentally validated is inherently limited. Even with predictive models to estimate and filter promoter strength, the results often represent a relative ranking, serving as an approximate correlation across the large population of generated promoters. This correlation, although robust at the scale of millions, does not guarantee that the strength distribution of a subset of hundreds of experimentally selected promoters will align with the predicted trends.

To address this limitation and further refine the selection process, this study employed a multi-model fusion and cross-validation strategy. By integrating predictions from a deep learning network and a RandomForestRegressor machine learning model, the pool of target promoters was narrowed significantly.

First, promoter strength was predicted separately using the deep learning and machine learning models. Promoters were then stratified into deciles based on predicted strength. Promoters that ranked in the top decile or quintile in both models were identified as high-confidence candidates, with a higher probability of exhibiting strong activity. Due to the limited overlap between the top-ranked predictions from the two models, this strategy effectively reduced the pool of candidates, significantly decreasing the experimental workload for validation.

Prior to the final fusion analysis, sequences similar to the target promoters were identified within the training dataset. These sequences, along with their associated strength values, were used to create a similarity-based subset. This subset was employed for the incremental training of the deep learning model, during which a portion of the original model’s parameters was randomly locked to ensure stability. This incremental training further enhanced the deep learning network’s ability to predict the strength of novel promoters with greater accuracy.

To validate the effectiveness of the models, 95 promoters that adhered to the predefined generation rules were selected using the GAN-based promoter generation method. The strength of these new mutant promoters was predicted using both the deep learning and machine learning models. Experimental validation was subsequently conducted using existing laboratory data, confirming the predicted strengths and demonstrating the utility of this integrated computational and experimental approach for promoter design and characterization.

### 2.2. Analysis of Quantile Hit Rates Across Models on the Test Set

Figure 2 presents the analysis of the quantile hit rates derived from the test data in the dataset using the deep learning prediction model. The results reveal that at lower ground truth quantile ranges (e.g., 10%, 20%), the model’s hit rates are typically low. However, as the predicted quantile increases (from 20% to 50%), the hit rates improve progressively at higher ground truth quantile ranges (e.g., 40%, 50%). This trend indicates that the model’s predictive capability strengthens as the prediction interval expands, with the performance of the fusion prediction model being particularly notable.

When analyzing the deep learning and machine learning models individually, their quantile hit rates increase with higher quantile ranges. The differences between the two models also become more pronounced as quantiles rise. At lower quantile ranges, such as 20%, the performance of the deep learning and machine learning models is comparable. However, for other quantiles, the hit rates of the deep learning and machine learning models diverge to varying degrees. This comparative analysis allows for an evaluation of each model’s effectiveness across different data ranges.

The figure also highlights the intersection of predictions from both models (i.e., the combined hit rate where predictions from both models align). This intersection typically indicates more precise matches, reflecting a high degree of agreement between model predictions and ground truth values. Because the intersection represents a narrower prediction range, its hit rate is often higher. This is particularly evident at higher quantile ranges (e.g., 30% and 50%), where the overlap between predictions from both models is more substantial.

Conversely, the union of the predictions from both models represents a broader coverage range. The union reflects the models’ ability to encompass ground truth values across a wider spectrum. However, due to the increased base size, the hit rate in the union region is generally lower. By combining the predictions from both models into a union, a greater range of ground truth values can be captured, but this broader coverage comes at the cost of reduced overall quantile hit rates.

In summary, notable differences are observed between the intersection and union of predictions generated by the two models. The intersection often enhances prediction accuracy, particularly at higher quantiles (e.g., 50%), where the fusion model (intersection) outperforms individual models. This underscores the effectiveness of model fusion as a strategy to improve prediction accuracy, especially when addressing prediction problems with complex distributions.

### 2.3. Analysis of Quantile Hit Rates Across Models on Out-of-Sample Mutant Promoters

Figure 3 presents the results for the out-of-sample mutant promoters generated by the framework. As the prediction quantile range increases from 20% to 50%, the quantile hit rate exhibits a clear upward trend, particularly in the intersection region, where the improvement is most pronounced. Individual models show varied performance across different prediction quantiles, with neither model consistently outperforming the other. This suggests that each model has distinct suitability depending on the data characteristics. However, this disparity diminishes in higher quantile ranges, indicating that the predictive performance of the two models converges at higher quantile intervals.

The fused predictions, especially those using the intersection strategy, are particularly effective in enhancing predictive performance, especially at higher quantile ranges. Across all quantile intervals, the hit rate of the intersection is consistently higher than that of the union, particularly at higher true value quantiles (e.g., 40% and 50%). This demonstrates that combining the predictions of both models using the intersection approach better captures the true value range, thereby improving prediction accuracy. These findings highlight that model fusion (via intersection) is an effective strategy for improving predictive capability in complex tasks.

Compared to the results in Figure 2, which focused on the test set, the performance of the two models is not entirely consistent across different datasets. The deep learning model is more suitable for the test set data, whereas the machine learning model performs better on out-of-sample mutant promoters. Overall, however, the intersection-based fusion of the two models demonstrates greater stability and improved performance across both datasets. Additionally, the comparison between the test set and mutant promoter results reveals that as the sample size increases, the models’ analytical performance becomes more robust and evident.

### 2.4. Hit Statistics Analysis of Each Model on the Test Set and Out-of-Sample Mutant Promoters

In practical experiments aimed at identifying or constructing novel promoters, the goal is not to work with hundreds or thousands of promoters but to find one or a few highly effective promoters tailored to a specific gene. Minimizing the experimental scope to identify high-efficiency target promoters is an effective strategy for accelerating experiments and reducing resource consumption. Conventional analyses often involve screening large datasets to identify statistical patterns, but the extensive sample size makes direct experimental implementation challenging. Therefore, to enhance experimental efficiency, it is crucial to narrow the range of samples to be analyzed while maintaining an acceptable level of accuracy.

Using the intersection of predictions from the two models allows for a significant reduction in experimental scope without compromising accuracy. As shown in Figure 4 and Figure 5, as the quantile range increases from the top two quantiles to the top five quantiles, the intersection between the two models’ predictions grows larger. This indicates that within broader predictive ranges, the predictions of the two models become more consistent. Higher quantiles effectively reduce the number of samples, and as the quantile range expands, the alignment between the model predictions and the ground truth values improves. The proportion of “non-intersection” samples decreases markedly, whereas the proportion of “ground truth hits” (e.g., 10%, 20%, 30%, etc.) increases. However, the added hits primarily correspond to lower ground truth value quantiles.

From the perspective of experimental workload and optimization, narrowing the quantile range for filtering (e.g., using only the top two quantiles) can significantly reduce experimental effort and computational resource consumption. Conversely, expanding the filtering range (e.g., to the top five quantiles) increases the number of predicted hits but also raises the computational workload.

## 3. Discussion

This study presents a systematic framework for the design and prediction of *Halomonas* bacterial promoters, combining generative adversarial networks (GANs) with multi-model predictive methods. By enhancing the accuracy of promoter strength prediction, this work introduces a novel tool for synthetic biology. The design and prediction framework not only improves the prediction of *Halomonas* bacterial promoters but also provides new insights for promoter design in other microbial systems. From our perspective, the integrated framework of promoter design and strength prediction proposed here can play a critical role in the following areas, including but not limited to the *Halomonas* bacteria domain. (1) Although this study focuses on *Halomonas* bacteria, the GAN-based framework can be extended to other microbial systems. By retraining the GAN model, specific promoter sequences can be designed for various species or genomic contexts. (2) By combining biological features such as the GC content and Moran’s coefficient with machine learning methods, the model achieves improved prediction accuracy while offering a novel analytical perspective for biological sequence design. (3) The proposed framework is highly scalable and can be integrated with automated experimental platforms, enabling an end-to-end workflow from design to experimental validation. (4) By integrating deep learning and machine learning models, this study successfully addresses predictive challenges in complex data scenarios. This strategy has the potential to be extended to other fields, such as genomics and proteomics.

Despite the success of this study, there are several limitations to address. First, the generative capacity of the GAN model relies on existing data, which may restrict the diversity of generated sequences. Incorporating promoter data from additional species or multi-omics datasets in the future could enhance sequence diversity. Second, significant differences exist between the high-dimensional feature extraction methods used in the deep learning model and the feature engineering approach employed in the random forest model. This inconsistency in feature space may increase the complexity of model fusion. Additionally, although the fused model outperforms individual models, its computational complexity is relatively high, posing challenges for practical application in resource-limited laboratory settings. Future efforts could focus on model compression or more efficient feature selection strategies to address this issue.

## 4. Materials and Methods

### 4.1. Dataset Construction

Due to the absence of an existing promoter database for *Halomonas* TD01, we first constructed a comprehensive database of *Halomonas* TD01 promoters and their corresponding strengths. Using the whole-genome sequences of *Halomonas* TD01 microorganisms, we selected 200 bp upstream of the ATG start codon of each gene as the core region for promoter screening. To identify promoter features such as the −10 box and −35 box, we employed BPROM (http://www.softberry.com/, accessed on 24 November 2024) to predict these elements for each candidate sequence. Promoters were defined as sequences extending 50 bp upstream from 10 bp beyond the predicted −10 box. Sequences lacking predictions for either the −10 box or −35 box (e.g., empty, zero, or NaN values) were excluded from the dataset. Given the relatively simple promoter structure in prokaryotes, which lacks complex regulatory networks and transcription factor interactions, the gene expression level is often a direct proxy for the strength of the corresponding promoter. Leveraging RNA-seq data obtained in our laboratory, we characterized promoter strength using the expression levels of associated genes. This process resulted in the construction of a *Halomonas* TD01 promoter database containing 3324 promoter sequences and their corresponding transcriptional strengths.

A high-quality promoter database is fundamental to constructing precise and efficient predictive models, particularly in studies requiring the accurate prediction and optimization of complex systems. The quality of the database directly determines the depth and breadth of research outcomes. *Escherichia coli*, as a well-studied model organism, boasts a database of over 10,000 known promoters, with promoter strengths quantified using dRNA-seq [27]. Previous studies in our laboratory have revealed significant structural and functional similarities between the promoters of *Halomonas* and *E. coli*. These shared features provide an opportunity to leverage the extensive knowledge and methodologies developed for *E. coli* in studying *Halomonas* promoters. However, *Halomonas* promoter sequences and their regulatory mechanisms are inherently more diverse and complex, reflecting the unique environmental adaptations of *Halomonas*. This complexity necessitates a nuanced approach that carefully considers both the similarities and differences to achieve accurate predictions and meaningful insights. Deep neural networks (DNNs), as employed in this study, require large datasets for effective training and convergence. Given the relatively limited data available for *Halomonas* promoters, we supplemented the training dataset with the *E. coli* promoter database. This augmentation not only increased the volume of training data but also facilitated the regression convergence of the model, thereby enhancing its predictive performance for *Halomonas* promoter strength.

### 4.2. Construction of the GAN Model for Promoter Generation

The generative adversarial network (GAN) was designed to learn the features of promoter sequences in the curated *Halomonas* promoter database. Since neural networks cannot directly process string inputs, the DNA sequences, composed of the nucleotide bases A, T, C, and G, were transformed using one-hot encoding. Each nucleotide was represented as a four-dimensional vector, enabling the sequences to be converted into one-hot encoded vectors suitable for neural network processing. The GAN architecture consists of two key components: the generator (Figure 6) and the discriminator (Figure 7). These two figures present the network architectures automatically generated using Python’s torchviz package (python 3.8), based on the code developed in this study. The generator network (Figure 6) receives a noise vector, typically randomly generated, as input. Using a long short-term memory (LSTM) network, the generator produces a DNA sequence in the form of one-hot encoded vectors. Each position in the sequence is represented as a four-dimensional vector, which corresponds to the probability distribution over the four nucleotide bases (A, T, C, G). The lengths of the promoters generated by the generator network were constrained using hyperparameters to match those of the promoters in the constructed dataset. The discriminator network (Figure 7) evaluates whether a given sequence is “real” (originating from the actual *Halomonas* promoter database) or “generated” (produced by the generator). Figure 7 also demonstrates that a portion of the discriminator’s input network architecture is composed of the generator. The discriminator accepts one-hot encoded DNA sequences as input and outputs a probability score, indicating the likelihood of the sequence being authentic.

The training process is divided into two parts: generator training and discriminator training. The generator improves its output by attempting to deceive the discriminator, making the generated sequences increasingly realistic. The loss function for the generator is the negative log probability of the discriminator classifying the generated sequence as “real”. The objective is to maximize the probability that the discriminator perceives its output as “real”. The discriminator calculates the probabilities for real samples and fake samples (generated by the generator) and aims to maximize its ability to classify real and fake sequences. The discriminator’s loss function is a binary cross-entropy loss, designed to optimize the model so that the output for real sequences approaches 1 and for fake sequences approaches 0. Through adversarial training, the generator and discriminator compete with each other and improve iteratively. The generator continuously enhances the quality of the generated sequences, whereas the discriminator becomes more adept at distinguishing between real and generated sequences.

Screening and optimization process: After each training iteration, the generator produces a batch of new DNA sequences. The code calculates the GC content (the proportion of G and C bases in the DNA sequence) and Moran’s coefficient (an autocorrelation index derived from dinucleotide frequencies) for each generated sequence. These metrics are used to evaluate the biological characteristics of the generated sequences. The generated sequences are screened based on the differences between their actual values and target GC content and Moran’s coefficient. Sequences that meet the preset conditions are retained. A tolerance threshold (gc_tolerance and moran_tolerance) allows for some deviation. The Moran’s coefficient helps evaluate the pattern and structural organization of the DNA sequences, whereas the GC content is a key physical property often used to analyze DNA stability and structure. Finally, the selected DNA sequences that meet the criteria are stored in a file for use in promoter strength prediction.

This GAN framework not only generates DNA sequences but also incorporates biological metrics such as the GC content and Moran’s coefficient into the screening process. This ensures that the generated DNA sequences have biological relevance and applicability.

### 4.3. Promoter Strength Prediction Model

The promoter strength prediction model integrates a deep learning framework composed of BiLSTM and CNN architectures with a machine learning approach based on random forest, achieving a fusion prediction strategy.

Construction of a deep learning-based prediction model 1 for DNA promoter strength using BiLSTM and CNN: Figure 8 illustrates the architecture of the first model, which is designed for predicting DNA promoter strength using a deep learning framework with cross-validation and multi-input features. The model leverages the convolutional processing of k-mer features and position-specific scoring matrix (PSSM) features to construct and train a hybrid architecture combining convolutional neural networks (CNNs) and bidirectional long short-term memory (BiLSTM) networks. BiLSTM is utilized for extracting features from k-mer data. Unlike conventional LSTM, BiLSTM captures both forward and backward dependencies within sequential data, making it particularly suitable for analyzing genomic sequences. For feature extraction, the k-mer and PSSM inputs are processed using one-dimensional and two-dimensional convolutional layers, respectively. The convolutional layers identify local patterns in the sequence, whereas the pooling layers reduce feature dimensions through down-sampling, minimizing the risk of overfitting.

The model architecture comprises two independent input branches: (1) k-mer features: The first input represents k-mer features encoded using Word2Vec. Each input sequence consists of 48 time steps with 100-dimensional features per step, resulting in an input dimension of (48, 100). (2) PSSM features: The second input branch processes PSSM data, represented as a 20 × 20 matrix with a single channel, i.e., an input dimension of (20, 20, 1). PSSMs encode the positional frequencies of nucleotides within a sequence, highlighting the conserved regions. The PSSM data are transformed into three-dimensional tensors and processed through convolutional layers to extract pattern features. Standardization is applied to both the DNA sequences and PSSM data to improve convergence during training and enhance model stability. Additionally, the target variable y (promoter strength) undergoes logarithmic standardization to facilitate model fitting.

For processing, the k-mer features are analyzed using a combination of CNN and BiLSTM networks, whereas the PSSM features are primarily handled through CNN layers. The processed features from both branches are concatenated into a unified feature vector, which is then passed through a fully connected layer with an output dimension of 512, utilizing a ReLU activation function. The final fully connected layer integrates the features from the two networks and outputs the predicted promoter strength.

The model employs mean squared error (MSE) as the loss function and is optimized using the Adam optimizer. A ReduceLROnPlateau callback is implemented to automatically reduce the learning rate if the validation loss plateaus, mitigating overfitting and addressing potential oscillations caused by an excessively high learning rate.

The model can be constructed de novo or fine-tuned through incremental training, which integrates principles of transfer learning. Incremental training allows the further optimization of a pre-existing model rather than initiating training from scratch. A notable feature of this approach is the option to “randomly lock specific layers”, where a selected proportion of model parameters are frozen during training. This ensures that previously acquired knowledge within the model is retained, avoiding the need for complete retraining while minimizing the risk of overfitting to new data. This strategy facilitates efficient model updates using minimal additional data, thereby improving training efficiency and adaptability.

Construction of a RandomForest-based machine learning prediction model 2: The second predictive model is built using the RandomForestRegressor algorithm. In the original dataset, promoter sequences are provided as DNA strings, which cannot be directly processed by machine learning algorithms. To address this, the input data are transformed into numerical feature matrices through various feature extraction techniques, such as CountVectorizer and TfidfVectorizer. These methods convert DNA sequences into numerical representations (X) suitable for machine learning. The extracted features (X) are then standardized, ensuring a mean of 0 and a variance of 1 for each feature, to enhance the stability of model training. Promoter strength, the target variable in this regression task, undergoes logarithmic transformation prior to training. This transformation aids the model in handling data with large variability, improving the fit across a wide range of values. To refine the input features, feature selection is performed. Methods such as SelectKBest, which selects the top 20% of features most strongly correlated with the target variable (or a user-defined number of features), and SelectFromModel, based on RandomForest, are utilized. These approaches help reduce dimensionality and retain only the most relevant features for prediction. Hyperparameter tuning is conducted using GridSearchCV, enabling the identification of the optimal parameter configuration for the RandomForestRegressor. Parameters such as n_estimators (number of trees), max_depth (maximum depth of each tree), and max_features (number of features considered for splitting) are systematically explored. During this process, cross-validation is employed to ensure robust model generalization and mitigate overfitting.

### 4.4. Feature Engineering Method

Chaos game representation: The chaos game representation (CGR) algorithm is a powerful technique for mapping sequence information into a two-dimensional space [28]. By iteratively plotting points based on the sequence of four nucleotides (A, T, C, G), CGR generates a two-dimensional image that captures the structural features of the DNA sequence. In the CGR framework, each nucleotide is assigned a specific coordinate in a two-dimensional plane: A = (0, 0), T = (1, 0), C = (0, 1), G = (1, 1). The algorithm begins at an initial point, and with each nucleotide in the sequence, the coordinates are updated recursively. At each step, the new coordinate is calculated as an interpolation between the current coordinate and the coordinate associated with the nucleotide being processed. This recursive process continues for the entire DNA sequence, ultimately producing a two-dimensional image matrix representing the sequence. The resulting image serves as both a visualization and analytical tool, enabling the identification of structural patterns and features within the DNA sequence. CGR is particularly useful for detecting local structural characteristics, assessing sequence complexity, and uncovering regularities in genomic sequences. This visualization technique has been widely employed in genomics and bioinformatics to reveal potential patterns within DNA sequences, contributing to deeper insights into genetic architecture.

For the *i*-th nucleotide ($b_{i}$), the coordinates are updated using the following recursive formula:(1)$$x_{i}=\frac{x_{i-1}+x_{b_{i}}}{2}$$
(2)$$y_{i}=\frac{y_{i-1}+y_{b_{i}}}{2}$$

Position-specific scoring matrix (PSSM): The position-specific scoring matrix (PSSM) is a matrix-based representation of DNA sequence features [29,30,31]. By leveraging the chaos game representation (CGR) of a sequence and applying an equally weighted transformation, PSSM quantifies the frequency of nucleotide pairs at specific positions within the sequence. This approach enables the functional analysis and alignment of sequences and provides insights into local sequence features and nucleotide preferences at individual positions. PSSM is widely employed in genomics to characterize key sequence features, including transcription factor binding sites and gene regulatory elements. For example, in the analysis of transcription factor binding sites, PSSM helps uncover the sequence preferences of specific transcription factors, elucidating their interaction with DNA.

The combination of CGR and PSSM methods effectively captures both the local and global features of DNA sequences, making them suitable for a wide range of genomic analyses. The visual representation offered by CGR facilitates intuitive sequence feature analysis, whereas PSSM provides a quantitative framework for characterizing these features. Each DNA sequence is first converted into its CGR representation, from which the corresponding PSSM is generated to describe the sequence’s structural and functional characteristics.

Features of the deep learning model: K-mer-based feature extraction and Word2Vec training. K-mer feature extraction [32,33,34]: DNA sequences, composed of four nucleotides (A, T, C, G), can be segmented into overlapping k-mers of a fixed length, typically *k* = 3. Each k-mer is treated as a “word” (e.g., GTA, TAG), and the overlapping nature of *k-mers* allows for the capture of local sequence features and conserved structures. By considering every consecutive triplet of nucleotides, *k-mer* extraction provides insights into short-sequence interactions, which are particularly relevant for analyzing genomic patterns. The 3-mer representation is a commonly used length due to its efficiency in capturing local sequence interactions while preserving manageable computational complexity.

*K-mer* feature extraction reduces the dimensionality of raw DNA sequence data, facilitating subsequent processing and analysis. This approach is widely used in genomics and proteomics for analyzing sequence motifs and functional elements:(3)$${Kmer}_{i}=b_{i}b_{i+1}b_{i+2}\;\;for\;i=1,2,\cdots ,n-2$$
where $b_{i}$ represents the *i*-th nucleotide in the DNA sequence, n denotes the total length of the sequence, and $b_{i}\in (A,T,C,G)$.

Word2Vec embedding [35,36]: Word2Vec is a machine learning technique that maps discrete “words” (in this case, *k-mers*) into continuous vector representations by learning from their contextual relationships within a sequence. Using the extracted *k-mer* sequences, a Word2Vec model is trained to generate vector embeddings for each *k-mer* [37]. This training leverages the semantic properties of *k-mers*, predicting the surrounding k-mers (context) based on a target *k-mer*, and vice versa. The resulting embeddings capture the inherent patterns and nucleotide combination preferences in the DNA sequence. These vector representations encode the local structural and functional features of the sequence, providing valuable inputs for biological analysis and machine learning tasks. By translating discrete *k-mers* into high-dimensional continuous vectors, Word2Vec embeddings reflect similarities between *k-mers*, enabling the discovery of hidden patterns and correlations within genomic data.

Machine learning model features: String-based feature engineering: Various feature extraction methods were employed to derive different representations from DNA promoter sequences, including classical text vectorization techniques (e.g., CountVectorizer, TF-IDF) and deep learning-based embedding methods (e.g., Word2Vec, BERT) [38,39]. The extracted features were then combined to form the input matrix for the machine learning model.

#### 4.4.1. String Feature Extraction Methods

CountVectorizer: Converts DNA sequences into numerical features, similar to the bag-of-words model, where each nucleotide (A, T, C, G) is treated as a feature.

CountVectorizer_n_grams: Similar to CountVectorizer but using n-grams (combinations of two or three consecutive nucleotides) to represent local structures within the sequence. This method captures local information, such as the continuity of certain nucleotide pairs, by counting occurrences of n-grams within the DNA sequence.

TfidfVectorizer: Extracts features by calculating the term frequency–inverse document frequency (TF-IDF). Although commonly used in text analysis, it can also be applied to DNA sequences to capture important patterns within the sequence.

HashingVectorizer: Maps DNA sequences into a fixed-dimensional feature space using a hash function without storing all features, making it highly efficient for large-scale data processing. This method is particularly useful when handling vast amounts of DNA sequence data.

LDA (latent Dirichlet allocation): A topic modeling approach traditionally used in text analysis, which decomposes sequences into multiple “topics”, where each topic is composed of several nucleotides.

LSA (latent semantic analysis): A dimensionality reduction technique using singular value decomposition (SVD) that extracts latent semantic features from sequences. Both LDA and LSA map DNA sequences into a latent topic space, helping to uncover hidden structural patterns within the sequences.

PCA (principal component analysis): A dimensionality reduction method used to extract the principal components that explain the most variance in the data. PCA simplifies high-dimensional data while retaining the most important features.

t-SNE (t-distributed stochastic neighbor embedding): A dimensionality reduction technique particularly useful for visualizing high-dimensional data by mapping it into two or three dimensions. Both PCA and t-SNE help simplify the feature space while preserving key sequence information and facilitating the visualization of high-dimensional data.

FastText: Similar to Word2Vec, but it considers subword (subsequence) information, allowing it to better handle rare or out-of-vocabulary terms.

Doc2Vec: Like Word2Vec, but it generates an embedding vector for entire sequences, making it suitable for generating fixed-length feature representations for the entire DNA sequence.

GloVe (global vectors for word representation): A word embedding method based on global co-occurrence statistics that maps each character in the sequence into a fixed-size vector space.

BERT (bidirectional encoder representations from transformers): A pre-trained model based on transformers that generates context-aware embeddings for each position within the sequence. By leveraging the bidirectional context, BERT captures more complex sequence patterns.

Deep learning-based methods like Word2Vec, FastText, Doc2Vec, GloVe, and BERT learn contextual information from the sequences and are capable of capturing complex patterns in DNA sequences. These methods are particularly effective in tasks such as predicting transcription factor binding sites and promoter activity.

The application of multiple feature extraction techniques enables the generation of multi-level, multidimensional feature representations from DNA promoter sequences, including local nucleotide pair frequencies and global latent semantic structures. These techniques range from traditional statistical approaches to modern deep learning methods, capturing a wide variety of patterns and relationships within the sequences. The extracted features are subsequently used for machine learning model training, enhancing the model’s generalization ability and predictive accuracy for promoter strength estimation.

#### 4.4.2. Machine Learning Model Features: Non-String-Based Feature Engineering

GC content: The *GC content* represents the proportion of guanine (G) and cytosine (C) bases in a DNA sequence. This feature is calculated by traversing all input DNA sequences and determining the *GC content* of each sequence individually [40]. The GC content provides insights into the relative abundance of G and C nucleotides in the sequence, which has significant implications for DNA stability, transcription efficiency, and other biological functions. The *GC content* is commonly used to analyze the structural and functional characteristics of DNA sequences. From a transcriptional and translational perspective, variations in the *GC content* can influence gene expression by affecting transcription and translation efficiency. Regions with higher *GC content* often harbor more transcription factor binding sites, potentially playing a role in gene regulation. From a genomic structure perspective, the *GC content* varies significantly across different regions of the genome. For instance, coding regions generally exhibit higher *GC content*, reflecting their functional and structural demands, whereas non-coding regions tend to have lower *GC content*.

The formula for calculating the *GC content* is expressed as follows:(4)$$GCContent=\frac{G+C\;bases}{Total\;bases}\times 100\%$$

Moran’s coefficient: Moran’s coefficient (Moran’s I) is utilized to calculate the dinucleotide Moran’s coefficient within a given set of DNA sequences [41]. This coefficient quantifies the spatial autocorrelation of specific features, such as dinucleotide frequencies, by examining the relationship between frequency values at one position and those at other positions within the sequence. It provides a measure of spatial dependency, offering insights into whether certain sequence characteristics exhibit spatial correlation.

In DNA sequences, spatial autocorrelation may reflect localized, stable, or spatially dependent properties associated with gene regulatory regions, gene functionality, or genomic features. For instance, genomic regions with high *GC content* often demonstrate elevated autocorrelation, as such regions are typically more stable and tend to cluster in specific genomic locations.

By calculating Moran’s coefficient for DNA sequences, researchers can discern patterns in base pair distributions within specific genes, regulatory elements, or other functional regions. For example, when analyzing genomes or transcription factor binding sites, Moran’s coefficient can reveal spatial preferences in the distribution of certain base pairs, thereby aiding investigations into gene regulation, chromatin architecture, and related genomic phenomena.

Moran’s coefficient is defined as follows:(5)$$I=\frac{n}{W}\cdot \frac{\sum_{i=1}^{n}\sum_{j=1}^{n}\omega_{ij}\left(x_{i}-\bar{x}\right)(x_{j}-\bar{x})}{\sum_{i=1}^{n}{\left(x_{i}-\bar{x}\right)}^{2}}$$
where $x_{i}$ and $x_{j}$ represent the dinucleotide frequency values, $\bar{x}$ denotes the mean of all frequency values, and $\omega_{ij}$ is the spatial weight between positions i and j within the sequence (typically assigned as 0 or 1, indicating whether a relationship exists between the positions).

Moran’s coefficient typically ranges from −1 to 1. A value of I = 1 indicates perfect positive spatial correlation, where dinucleotide combinations at adjacent positions are highly similar. Conversely, I = −1 signifies complete negative spatial correlation, where adjacent positions exhibit entirely different dinucleotide combinations. A value of I = 0 suggests no spatial correlation.

CKSNAP features: The CKSNAP (k-space nucleotide adjacent pair) features are derived from the frequency of dinucleotide pairs within DNA sequences [42]. By calculating the frequencies of nucleotide pairs separated by varying distances k, this method captures the relative distribution patterns of dinucleotide pairs in DNA sequences. The primary objective is to reveal the relative frequencies of nucleotide pairs within the sequence.

Different k-values correspond to varying nucleotide pair spacings, enabling the detection of distinct local structural characteristics within DNA sequences. Smaller k-values reflect relationships between closely adjacent nucleotide pairs, whereas larger k-values uncover interactions between more distantly spaced pairs.

By analyzing the frequencies of all 16 possible dinucleotide pairs (e.g., “AA”, “AT”, “GC”, “TG”), CKSNAP features effectively unveil the compositional and structural characteristics of DNA sequences. Certain nucleotide pairs may occur more frequently in specific genomic regions, such as gene coding areas or regulatory elements.

Given a DNA sequence $S=(s_{1},s_{2},\cdots ,s_{N})$, where $s_{i}\in (A,T,C,G)$ represents the nucleotide at position iii, a nucleotide pair $\left(s_{i},s_{i+k+1}\right)$ is defined for a given spacing k. Here, i indicates the position of the nucleotide, and i∈[1,N−k−1]. The spacing k represents the distance between the two nucleotides in the pair.

For a given DNA sequence, the frequency of each pair $\left(s_{i},s_{i+k+1}\right)$ is computed by counting its occurrences within the sequence. Denoting the occurrence count of a nucleotide pair $\left(s_{i},s_{i+k+1}\right)$ as $count(s_{i},s_{i+k+1})$, the total number of valid nucleotide pairs in the sequence is N−k−1, where N is the sequence length.

The frequencies of all nucleotide pairs are normalized to yield the occurrence frequency $F_{xy}(k)$, where x,y∈(A,T,C,G), representing the normalized frequency of the nucleotide pair in the sequence. The formula is expressed as follows:(6)$$F_{xy}(k)=\frac{count(x,y,k)}{N-k-1}$$
where $F_{xy}(k)$ represents the occurrence frequency of the nucleotide pair (x,y) at a spacing of k.

For a given k, the frequencies of all 16 possible nucleotide pairs (AA, AC, AG, AT, CA, CC, CG, CT, GA, GC, GG, GT, TA, TC, TG, TT) are calculated. These frequencies collectively form a *CKSNAP* feature vector, expressed as follows:(7)$$CKSNAP\left(k\right)=(F_{AA}\left(k\right),\;F_{AC}\left(k\right),F_{AG}\left(k\right),F_{AT}\left(k\right),\cdots ,F_{TT}(k))$$

For all spacings $k\in (0,1,\cdots ,K_{max})$, the feature vectors corresponding to each k can be concatenated to form a comprehensive *CKSNAP* feature vector:(8)$${CKSNAP}_{total}=(CKSNAP\left(0\right),\;CKSNAP\left(1\right),\cdots ,CKSNAP\left(K_{max}-1\right))$$

These features capture the local structural characteristics, repetitive patterns, and interactions between nucleotide pairs within DNA sequences. They also reflect preferences in transcription factor binding sites, the structural properties of gene regulatory regions, and other genomic attributes. Such features can be leveraged for subsequent bioinformatics analyses, including pattern recognition and other bioinformatics analyses.

EIIP Features: EIIP features are derived from a combination of EIIP values and k-mer frequencies [43,44]. EIIP (electron–ion interaction potential) represents an energy value that describes the interaction between nucleotides and electron–ion systems. This parameter characterizes nucleotide pairs based on the differences in their electron–ion interaction energy. Each nucleotide (A, C, G, T) is assigned a specific EIIP value, reflecting the unique electronic properties of the bases within a biological system. By calculating the frequency of each k-mer (subsequence of nucleotides) within a DNA sequence and multiplying it by its corresponding EIIP value, a feature vector is constructed. This vector captures both the local nucleotide structural features and their electronic properties. These features can be employed in various bioinformatics tasks, including pattern recognition, sequence classification, genomics analysis, and gene expression studies.

For a given sequence $S=(s_{1},s_{2},,s_{N})$ with a fixed k-value, where $s_{i}\in (A,T,C,G)$, the frequency of each k-mer is calculated as follows:(9)$$F_{pqr}(k)=\frac{count(pqr,k)}{N-k+1}$$
where count(pqr,k) represents the number of occurrences of the k-mer pqr in the sequence S, N denotes the length of the sequence, and $F_{pqr}(k)$ is the normalized frequency of the *k*-mer pqr in the sequence. Each nucleotide (A,T,C,G) is associated with a specific *EIIP* value. For any *k*-mer pqr, its *EIIP* value is calculated as the sum of the *EIIP* values of the nucleotides comprising the *k*-mer:(10)$$EIIP\left(pqr\right)=EIIP\left(p\right)+EIIP\left(q\right)+EIIP(r)$$
where pqr∈(A,T,C,G) represents the first, second, and third nucleotides of the *k*-mer.

After obtaining the frequency $F_{pqr}(k)$ and the *EIIP* value $EIIP\left(pqr\right)$ for each *k*-mer, the final feature value can be calculated as follows:(11)$$Feature(pqr,k)=F_{pqr}(k)\times EIIP\left(pqr\right)$$

For a given DNA sequence S and all possible *k*-values (ranging from 1 to the maximum *k*-value), the feature values for all *k*-mers can be calculated and combined into a comprehensive feature vector:(12)$$F_{s}=(Feature\left({pqr}_{1},k_{1}\right),Feature\left({pqr}_{2},k_{1}\right),\;Feature({pqr}_{n},k_{j}))$$
where ${pqr}_{n}$ represents all possible *k*-mers, $k_{j}$ denotes a specific *k*-value, and n indicates the total number of possible *k*-mers. This process results in a feature vector encompassing multiple characteristics derived from the sequence.

### 4.5. Strains and Cultivation

The *Escherichia coli* S17-1 strain was used for plasmid cloning and as a donor strain for conjugative transfer to *Halomonas* TD01. *Halomonas* TD01 was utilized for characterizing the strength of the *porin* promoter library. *E. coli* was cultured in LB medium (10 g/L of peptone, 5 g/L of yeast extract, 10 g/L of NaCl) at 200 rpm and 37 °C, with a pH of 7. The conjugation medium, LB20 (10 g/L of peptone, 5 g/L of yeast extract, 20 g/L of NaCl), was used to transfer the constructed plasmids from *E. coli* S17-1 to *Halomonas* TD01 via conjugation. *Halomonas* TD01, an extremophilic microorganism isolated from Ayding Lake, Xinjiang, was cultured at 200 rpm and 37 °C in LB60 medium, which is identical in composition to LB medium except for a NaCl concentration of 60 g/L. The pH of the medium was maintained between 8 and 9. Peptone and yeast extract were purchased from Oxoid Ltd. (Beixing Stoke, UK), and NaCl was procured from China National Pharmaceutical Group Corporation (Beijing, China).

In *Halomonas* TD01, the *porin* promoter drives the high expression of porin proteins and serves as a constitutive promoter for heterologous protein overexpression. The broadly active *porin* promoter library used in this study was constructed in-house by saturating mutations within the −10 box (four nucleotides) and the three nucleotides upstream of the −10 box of the wild-type *porin* promoter. Promoters from the *porin* promoter library were assembled into a pSEVA321 plasmid containing sfGFP via Gibson assembly. The resulting *pSEVA321-Pporin library-sfGFP* plasmids, constructed in *E. coli* S17-1, were transferred to *Halomonas* TD01 using conjugative transfer. The expression level of each *porin* promoter was characterized by fluorescence intensity (FI). Single colonies of *Halomonas* TD01 containing the constructed *porin* promoters were streaked on LB60 agar plates and incubated at 37 °C for 2 days. Individual colonies were picked and cultured in 96-well deep-well plates, with a culture volume of 1 mL at 37 °C and 1000 rpm. The cells were collected, resuspended in 0.5% PBS (phosphate-buffered saline), and analyzed for fluorescence. Fluorescence intensity (FI) was measured using an LSRII flow cytometer (BD Biosciences, Franklin Lakes, NJ, USA) and used to characterize the strength of each promoter in the *porin* promoter library [45].

## 5. Conclusions

This study addresses the challenges of limited data and the complexity of design and prediction in *Halomonas* bacterial promoter research by developing an innovative promoter generation and prediction framework. It highlights the broad applicability of artificial intelligence in synthetic biology. By establishing the first database of *Halomonas* bacterial promoters and their strengths, this study fills a significant gap in *Halomonas* bacterial research, providing essential data support for promoter design. The GAN-based promoter generation model leverages features such as the GC content and Moran’s coefficient to generate biologically plausible promoter sequences at scale. Using a feedback mechanism, the model optimizes the generated sequences, showcasing the advantages of generative adversarial networks in DNA sequence generation tasks. For promoter strength prediction, the proposed multi-model fusion strategy integrates the strengths of deep learning and machine learning. The deep learning model employs bidirectional long short-term memory (BiLSTM) and convolutional neural networks (CNNs) to capture sequence features, whereas the machine learning model enriches feature representation through string-based feature engineering (e.g., CountVectorizer, TF-IDF) and non-string-based feature engineering (e.g., GC content, Moran’s coefficient, and CKSNAP features). The predictions from both models are combined using an intersection-based strategy, significantly improving quantile hit rates and effectively narrowing the scope of experimental validation. On both the test set and the out-of-sample mutant promoter data, the fusion model outperformed the individual models, demonstrating its generalizability and robustness. Additionally, the functionality of the generated promoters was experimentally validated, confirming the practical utility of the generation and prediction framework. The experimental results indicate that the multi-model fusion strategy enhances prediction accuracy and efficiency, particularly in resource-limited settings, by reducing the experimental validation workload. By integrating deep learning-based graph features with diverse features from machine learning, this study achieves the multi-level modeling of promoter strength, providing new tools and insights for the design of gene regulatory elements.

Overall, the proposed promoter design and prediction framework not only advances the study of *Halomonas* bacterial promoters but also offers a viable strategy for gene regulation research in other species. The combination of GAN-based generation and multi-model fusion methods holds great promise for applications in microbial gene editing, synthetic pathway optimization, and industrial fermentation within synthetic biology. This work provides a novel approach to the intersection of artificial intelligence and life sciences. Future efforts could focus on optimizing the generation model and prediction framework, extending their applicability to broader genomic regions, and driving gene regulation research in extremophiles to new dimensions of exploration.

## Figures and Tables

**Figure 1 ijms-25-13137-f001:**
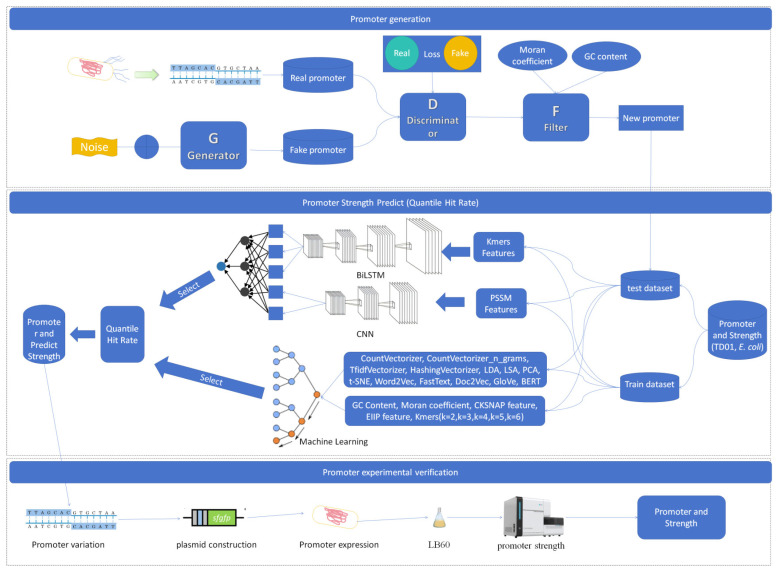
Workflow for the design, prediction, and validation of *Halomonas* promoters.

**Figure 2 ijms-25-13137-f002:**
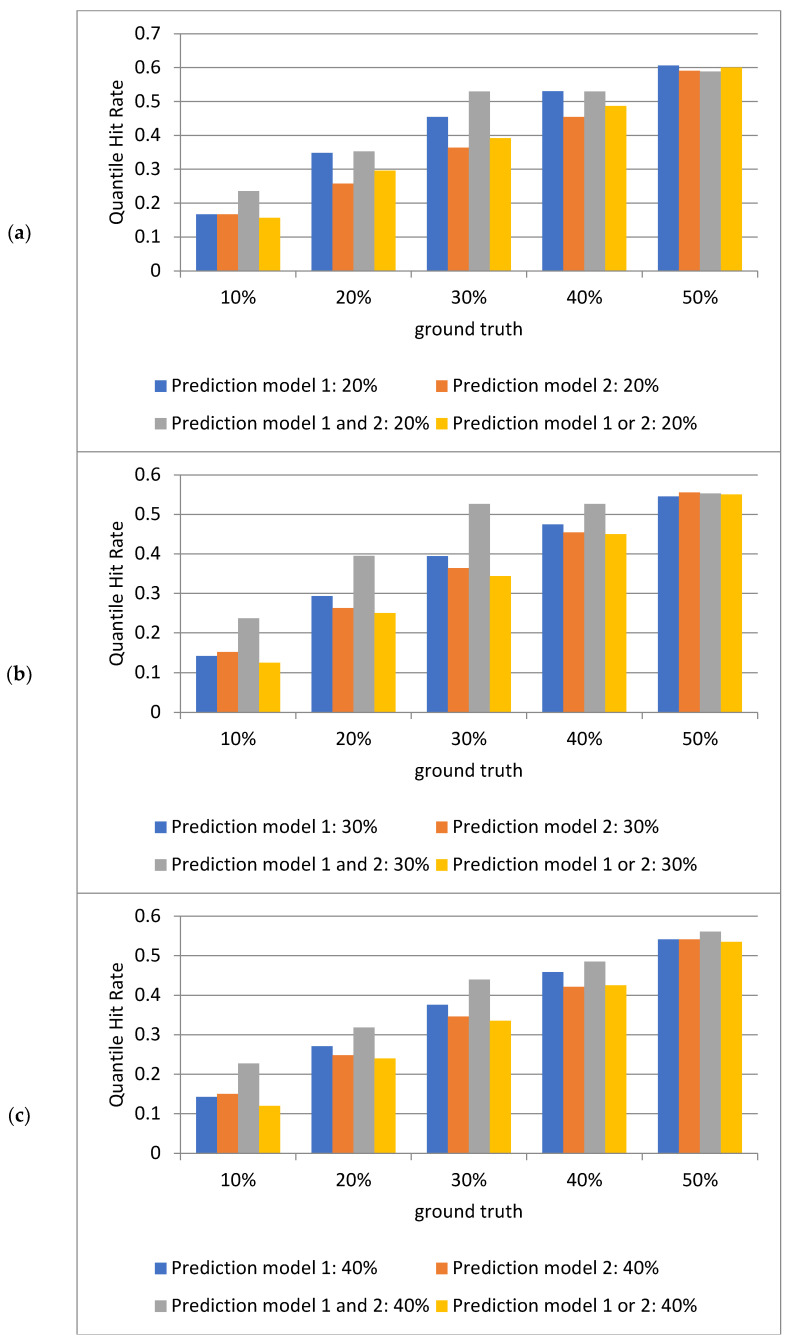
Analysis of the quantile hit rates (QHR) for the test set using deep learning (model 1) and machine learning (model 2) models. Each subfigure depicts the quantile hit rates for individual models, their intersection, and their union across various quantiles (20%, 30%, 40%, 50%). A higher hit rate indicates the greater accuracy of the model in predicting ground truth values within the respective intervals. (**a**) Hit rate statistics for the top two quantiles of the predicted values from both models compared to the 1st to 5th ground truth quantiles. (**b**) Hit rate statistics for the top three quantiles of the predicted values from both models compared to the 1st to 5th ground truth quantiles. (**c**) Hit rate statistics for the top four quantiles of the predicted values from both models compared to the 1st to 5th ground truth quantiles. (**d**) Hit rate statistics for the top five quantiles of the predicted values from both models compared to the 1st to 5th ground truth quantiles. Each subfigure contains five groups of bars, with four bars per group. The first bar in each group represents the percentage of hits where the deep learning model’s predictions align with ground truth values across quantiles. The second bar represents the percentage of hits where the machine learning model’s predictions align with ground truth values across quantiles. The third bar represents the percentage of hits where the intersection of predictions from both models aligns with ground truth values across quantiles. The fourth bar represents the percentage of hits where the union of predictions from both models aligns with ground truth values across quantiles. The horizontal axis in each subfigure denotes the ground truth value quantiles (1st to 5th quantiles corresponding to 10%, 20%, 30%, 40%, and 50%). The vertical axis represents the percentage of quantile hit rates.

**Figure 3 ijms-25-13137-f003:**
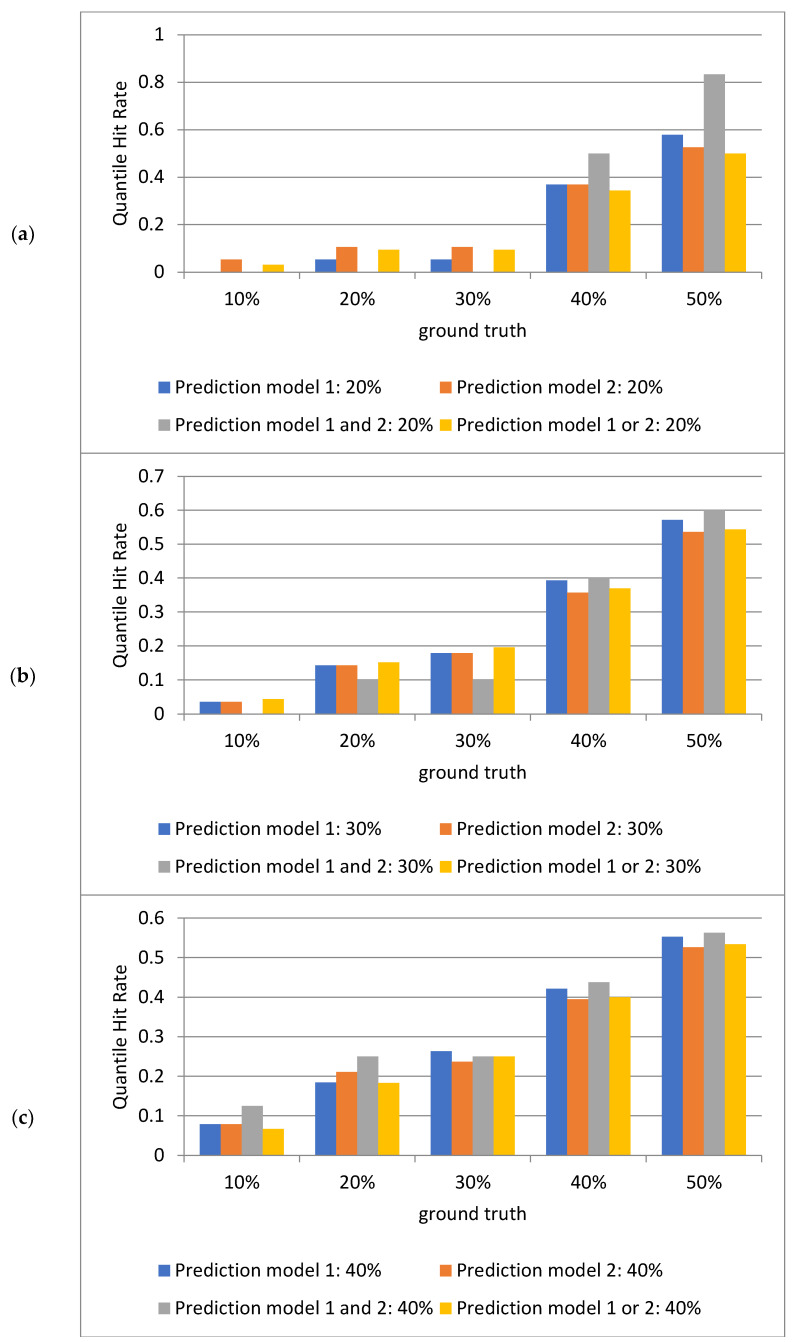
Analysis of the quantile hit rates (QHR) for out-of-sample mutant promoters using deep learning (model 1) and machine learning (model 2) models. Each subfigure depicts the quantile hit rates for individual models, their intersection, and their union across various quantiles (20%, 30%, 40%, 50%). A higher hit rate indicates the greater accuracy of the model in predicting ground truth values within the respective intervals. (**a**) Hit rate statistics for the top two quantiles of the predicted values from both models compared to the 1st to 5th ground truth quantiles. (**b**) Hit rate statistics for the top three quantiles of the predicted values from both models compared to the 1st to 5th ground truth quantiles. (**c**) Hit rate statistics for the top four quantiles of the predicted values from both models compared to the 1st to 5th ground truth quantiles. (**d**) Hit rate statistics for the top five quantiles of the predicted values from both models compared to the 1st to 5th ground truth quantiles. Each subfigure contains five groups of bars, with four bars per group. The first bar in each group represents the percentage of hits where the deep learning model’s predictions align with ground truth values across quantiles. The second bar represents the percentage of hits where the machine learning model’s predictions align with ground truth values across quantiles. The third bar represents the percentage of hits where the intersection of predictions from both models aligns with ground truth values across quantiles. The fourth bar represents the percentage of hits where the union of predictions from both models aligns with ground truth values across quantiles. The horizontal axis in each subfigure denotes the ground truth value quantiles (1st to 5th quantiles corresponding to 10%, 20%, 30%, 40%, and 50%). The vertical axis represents the percentage of quantile hit rates.

**Figure 4 ijms-25-13137-f004:**
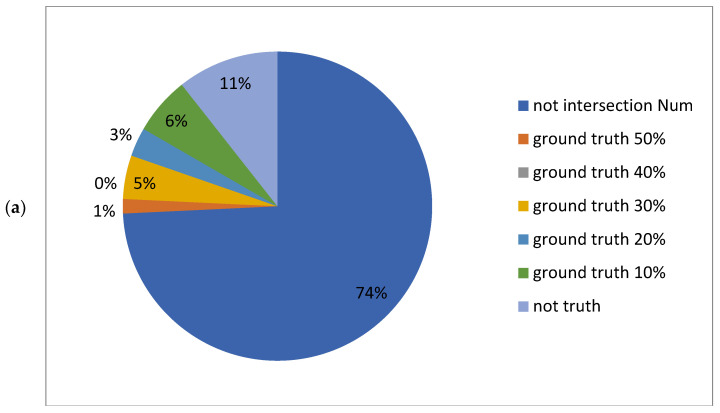
An analysis of the test set comparing the number of candidates and ground truth hits before and after intersecting the predictions of the deep learning and machine learning models. Each subfigure illustrates the candidate range for the two models at different quantiles (20%, 30%, 40%, 50%), the intersected candidate range, and the number of ground truth value hits within these ranges. A smaller candidate range implies a reduced experimental validation workload. (**a**) Statistics for the number of candidates, intersection size, and hits across the 1st to 5th quantiles of the ground truth values for the top two quantiles of the predicted values from both models. (**b**) Statistics for the number of candidates, intersection size, and hits across the 1st to 5th quantiles of the ground truth values for the top three quantiles of the predicted values from both models. (**c**) Statistics for the number of candidates, intersection size, and hits across the 1st to 5th quantiles of the ground truth values for the top four quantiles of the predicted values from both models. (**d**) Statistics for the number of candidates, intersection size, and hits across the 1st to 5th quantiles of the ground truth values for the top five quantiles of the predicted values from both models. In each subfigure, the entire pie chart represents the total candidate range initially screened by the two models, which includes both the intersected and non-intersected regions. The intersected region is further divided into sections representing the number of ground truth value hits across the five quantiles (10%, 20%, 30%, 40%, 50%) and the number of non-hits within the intersection. This visualization demonstrates how intersecting the models’ predictions reduces the candidate range while retaining or improving the proportion of ground truth value hits, thus optimizing experimental validation efforts.

**Figure 5 ijms-25-13137-f005:**
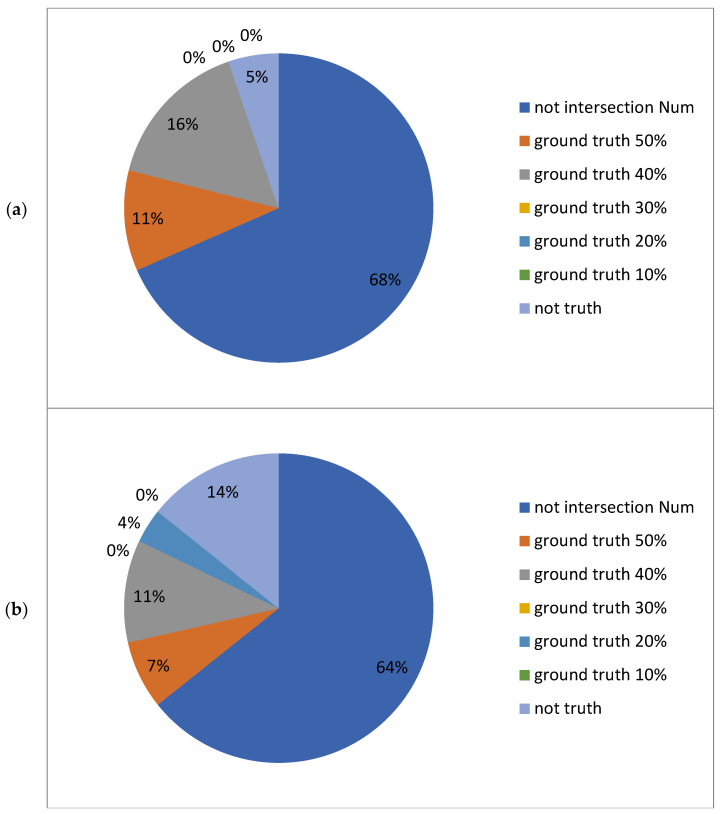
An analysis of the out-of-sample mutant promoters comparing the number of candidates and the ground truth hits before and after intersecting the predictions of the deep learning and machine learning models. Each subfigure illustrates the candidate range for the two models at different quantiles (20%, 30%, 40%, 50%), the intersected candidate range, and the number of ground truth value hits within these ranges. A smaller candidate range implies a reduced experimental validation workload. (**a**) Statistics for the number of candidates, intersection size, and hits across the 1st to 5th quantiles of the ground truth values for the top two quantiles of the predicted values from both models. (**b**) Statistics for the number of candidates, intersection size, and hits across the 1st to 5th quantiles of the ground truth values for the top three quantiles of the predicted values from both models. (**c**) Statistics for the number of candidates, intersection size, and hits across the 1st to 5th quantiles of the ground truth values for the top four quantiles of the predicted values from both models. (**d**) Statistics for the number of candidates, intersection size, and hits across the 1st to 5th quantiles of the ground truth values for the top five quantiles of the predicted values from both models. In each subfigure, the entire pie chart represents the total candidate range initially screened by the two models, which includes both the intersected and non-intersected regions. The intersected region is further divided into sections representing the number of ground truth value hits across the five quantiles (10%, 20%, 30%, 40%, 50%) and the number of non-hits within the intersection. This visualization demonstrates how intersecting the models’ predictions reduces the candidate range while retaining or improving the proportion of ground truth value hits, thus optimizing experimental validation efforts.

**Figure 6 ijms-25-13137-f006:**
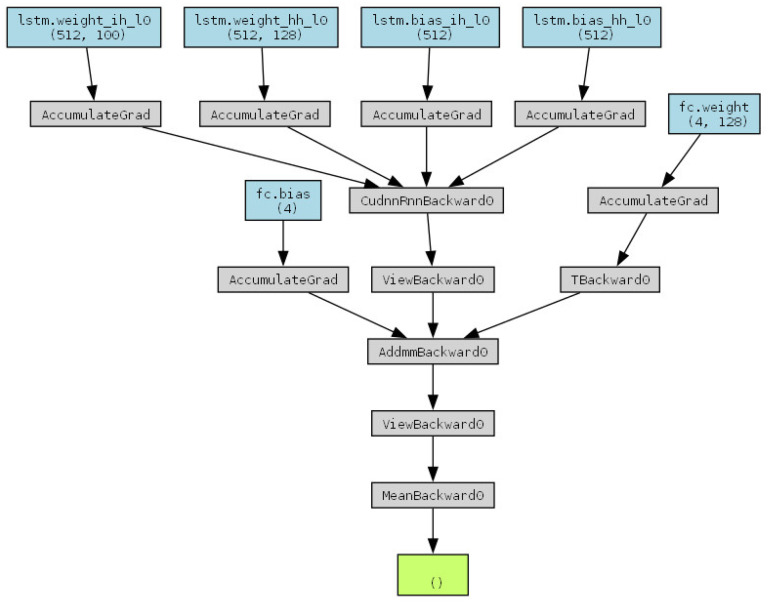
Generator network structure of the GAN model (generator_structure).

**Figure 7 ijms-25-13137-f007:**
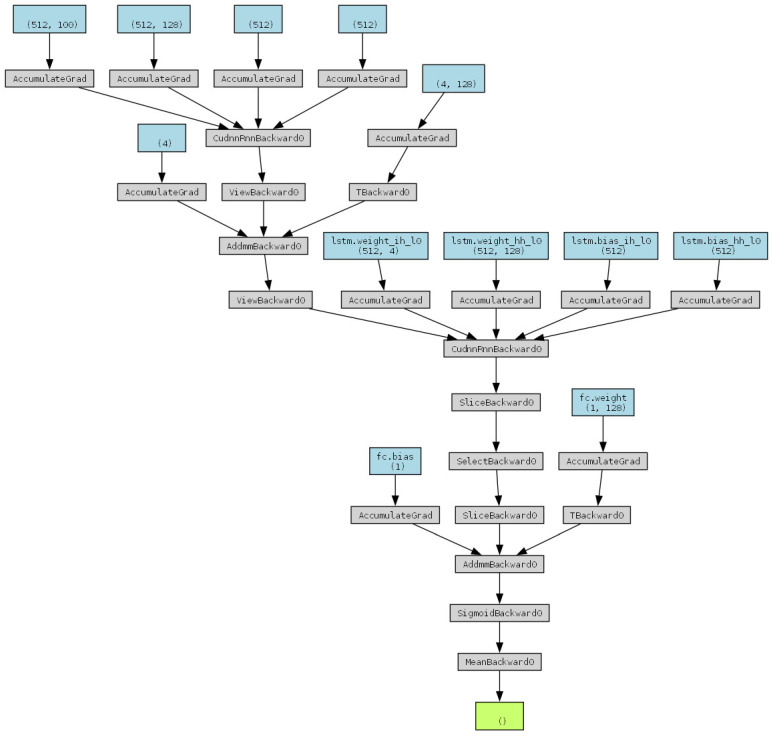
Discriminator network structure of the GAN model (discriminator_structure).

**Figure 8 ijms-25-13137-f008:**
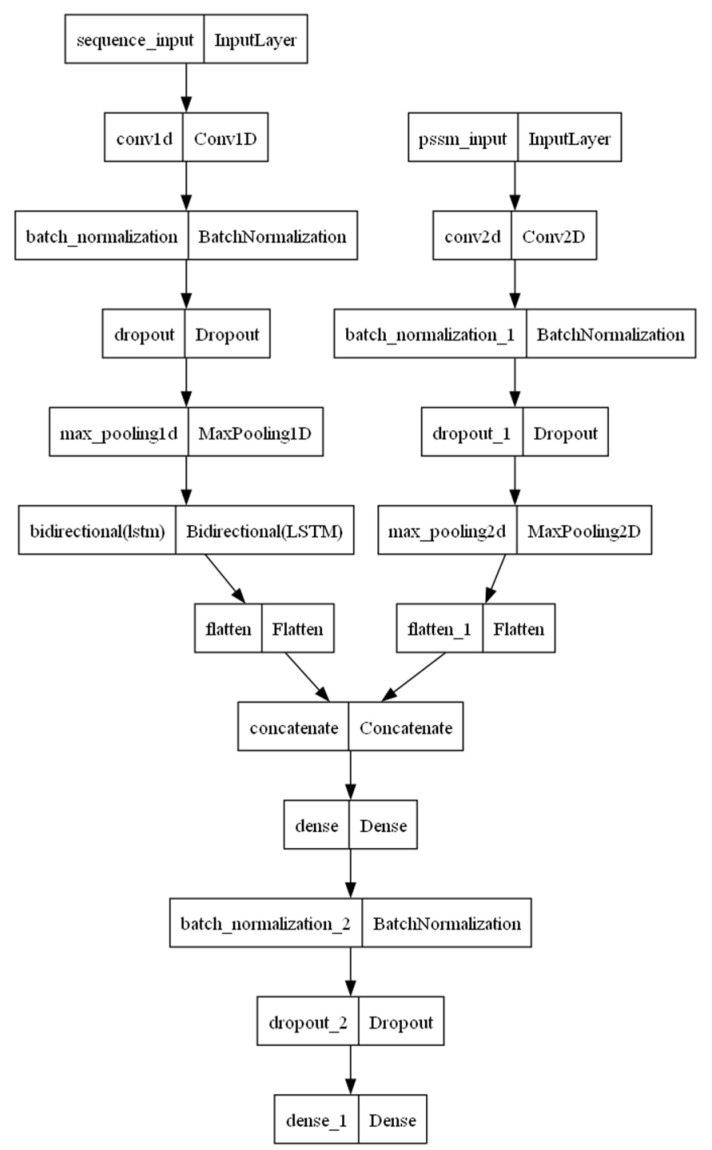
Prediction model 1: The model utilizes BiLSTM and CNN architectures to independently analyze the k-mer features and PSSM features of the input DNA sequences, respectively. In this model, BiLSTM and CNN operate as two distinct networks, where the BiLSTM component processes the k-mer features derived from DNA and the CNN component handles the PSSM features. The outputs of both networks are subsequently integrated through a fully connected layer for final prediction.

## Data Availability

The source code is publicly available at https://github.com/GBCRPromoter/GBCRPromoter (accessed on 24 November 2024).

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
