# Peer review of "Exploring the Promoter Generation and Prediction of Halomonas spp. Based on GAN and Multi-Model Fusion Methods"

_ijms, 2024, doi:10.3390/ijms252313137_

Round 1

Reviewer 1 Report

Comments and Suggestions for Authors

Promoters play an important role in an important role in genetic engineering and synthetic biology. In their work, the authors develop a new a multi-model fusion framework integrating deep learning and machine learning to enhance their prediction accuracy. Furthermore, the robustness and applicability of their model have been validated on diverse datasets. Overall, their work shows interesting contents. Suggesting that this work can be accepted after minor revision.

1. On their workflow for the design, prediction, and validation of halomonas promoters, the authors had better provide more detailed description.

2. Figure 6 and Figure 7 include rich contents, but the description is simple. The authors should provide more details so that their work can be better understood.

Author Response

1. Summary

We are grateful to the editor and the reviewers for their efforts to handle and review our paper. All the comments have been addressed and any changes to the manuscript are highlighted in yellow for clarity.

2. Point-by-point response to Comments and Suggestions for Authors

Comments 1: On their workflow for the design, prediction, and validation of halomonas promoters, the authors had better provide more detailed description.

Response 1: Thank you for your suggestion. To provide better clarity and explanation, we have revised and added the following content to the introduction:“The overall framework, depicted in Figure 1, comprises three key components: promoter design, strength prediction, and validation. Initially, a promoter strength database for Halomonas species was constructed. In the promoter generation phase, promoters were synthesized using a Generative Adversarial Network (GAN), wherein the GAN generator was responsible for producing noisy promoter sequences, and the discriminator evaluated whether these promoters met the target specifications by cal-culating the loss between the noisy and real promoters. Subsequently, the promoters generated by the discriminator underwent assessment of their biological characteris-tics, including the Moran coefficient and GC content. For the promoter prediction segment, a fusion predictive model integrating two distinct categories of models was employed. Firstly, deep learning models were trained on promoter datasets from both Halomonas and E.coli using k-mer and Position-Specific Scoring Matrix (PSSM) features, respectively. This deep learning models comprised two separate networks: Bidirection-al Long Short-Term Memory (BiLSTM) and Convolutional Neural Network (CNN). Concurrently, machine learning models, specifically Random Forests, were trained on the Halomonas promoter dataset utilizing both string-based and non-string-based feature engineering techniques. The fusion of these two modeling approaches constituted the proposed promoter strength prediction framework. Finally, the promoters generated by the GAN were subjected to rigorous screening, and the strengths of 95 novel mutant promoters were verified by experimental data. Comparative analyses revealed that the fusion model significantly enhanced the quantile hit rates in promoter strength predictions, underscoring its efficacy in accurately forecasting promoter activity.”

Comments 2: Figure 6 and Figure 7 include rich contents, but the description is simple. The authors should provide more details so that their work can be better understood.

Response 2: Thank you for your suggestion. Figures 6 and Figures 7 were automatically generated using Python’s torchviz package based on the code developed in this study. They primarily illustrate the overall architecture of the GAN network. For specific parameter settings, please refer to the code provided herein. In this study, the noise hyperparameter for the generator was set to 100, and the hyperparameter for promoter sequence length was configured to match the promoter lengths in the input database (i.e., 50 bp). Additionally, we have made the following additions and modifications to the original manuscript. To provide better clarity and explanation, we have revised and added the following content to Section 4.2::“The GAN architecture consists of two key components: the Generator (Figure 6) and the Discriminator (Figure 7). These two figures present the network architectures automatically generated using Python’s torchviz package, based on the code developed in this study. The generator network (Figure 6) receives a noise vector, typically randomly generated, as input. Using a Long Short-Term Memory (LSTM) network, the generator produces a DNA sequence in the form of one-hot encoded vectors. Each position in the sequence is represented as a four-dimensional vector, which corresponds to the probability distribution over the four nucleotide bases (A, T, C, G). The lengths of promoters generated by the generator network were constrained using hyperparameters to match those of the promoters in the constructed dataset. The discriminator network (Figure 7) evaluates whether a given sequence is “real” (originating from the actual Halomonas promoter database) or “generated” (produced by the generator). Figure 7 also demonstrates that a portion of the discriminator’s input network architecture is composed of the generator. The discriminator accepts one-hot encoded DNA sequences as input and outputs a probability score, indicating the likelihood of the sequence being authentic.”

Reviewer 2 Report

Comments and Suggestions for Authors

Zhao and colleagues designed a multi-model computational approach for designing and predicting gene promoters in Halomonas. It has been shown that this novel toolkit accurately predicts promotors and efficiently downsizes the candidate pool for downstream validation. Overall, I believe this is a publishable work, and I would appreciate it if the authors could address my questions following:

Is there anything special about the promotors of Halomonas? There is only one sentence that briefly introduces Halomonas as an extremophilic microorganism with ‘unique genomic architecture’, but it still does not justify the significance of studying such a bacterium for promotor designing and prediction. Can the authors include more information (preferably a dedicated paragraph) on the biology of this bacteria in the introduction?

In the second paragraph of page 4, the authors state that the deep learning model performed better since it ‘leverages high-dimensional features derived from graph-based information and graph convolution’, but it is fed by k-mer features and PSSM matrix features which are sequence-based, correct? If so, how can the deep learning model use graph information?

Figure 2 does not specify which model (1 vs 2) represents deep learning or machine learning. Also, how can one tell that the differences between these two models are negligible or significant? Is there a way to calculate p values or any other alternatives based on the results obtained from repeated runs? It is especially important when the authors state that ‘the differences between the intersection and union of predictions from the two models are striking’, which, to me, are not visually ‘striking’. Figure 3 shares these issues as well.

Author Response

Summary

We are grateful to the editor and the reviewers for their efforts to handle and review our paper. All the comments have been addressed and any changes to the manuscript are highlighted in yellow for clarity.

Point-by-point response to Comments and Suggestions for Authors

Comments 1: Is there anything special about the promotors of Halomonas? There is only one sentence that briefly introduces Halomonas as an extremophilic microorganism with ‘unique genomic architecture’, but it still does not justify the significance of studying such a bacterium for promotor designing and prediction. Can the authors include more information (preferably a dedicated paragraph) on the biology of this bacteria in the introduction?

Response 1: Thank you for your suggestion. Halomonas is an extremophile microorganism, and its extreme salinophilicity determines that it has a genome structure that is different from that of common microbial chassis bacteria.

In order to provide a clearer explanation, we added the following in the introduction: “Halomonas is a class of Gram-negative, salinophilic, alkaliphilic, rod-shaped bacteria that can survive at a salt concentration of 60 g/L and a pH of 8-9. It as a non-modal extremophile microorganism, has been widely used in the fields of biomaterials (e.g., PHA), biosurfactants and other fields. In addition, its salt tolerance gives it a significant competitive advantage in open fermentation systems to reduce production costs. What is special about the promoters of Halomonas is that they must be adapted to the high salt and alkaline environment in which the host cells are found. Promoters are DNA sequences upstream of genes that are responsible for regulating the initiation of gene expression. In Halomonas, the promoter core regions (-10 and -35 regions) typically retain sequence features similar to those of other Gram-negative bacteria, but because they are required to ensure normal gene expression under high-salt and high-alkaline conditions, Halomonas promoters may contain unique gene structures to be able to drive higher levels of gene expression under high osmolarity conditions. Therefore, studying Halomonas promoters and their stability in complex environments could optimize the expression of key enzymes, enhance the yield of targeted metabolites, and provide a broad scope for the development of novel bioproducts and green industrial production processes."

Comments 2: In the second paragraph of page 4, the authors state that the deep learning model performed better since it ‘leverages high-dimensional features derived from graph-based information and graph convolution’, but it is fed by k-mer features and PSSM matrix features which are sequence-based, correct? If so, how can the deep learning model use graph information?

Response 2: Yes, deep learning employs convolutional networks to process image-like data. In this study, the k-mer features possess dimensions of (2991,48,100), and the Position-Specific Scoring Matrix (PSSM) features have dimensions of (2991,20,20,1). These feature dimensions extend beyond conventional two-dimensional matrices, representing high-dimensional characteristics. For example, the k-mer features with dimensions (2991,48,100) can be understood from a macro perspective as  2,991 images, each with a size of (48,100). The deep learning network applies convolutional operations at each stride to extract information from each image. In contrast, traditional machine learning approaches utilize two-dimensional features, where row indices correspond to data samples and column indices correspond to feature attributes. Because deep learning can handle higher dimensional features, it captures a more comprehensive range of information.

To provide better clarity and explanation, we have added the following content to Section 2.1: “It is important to note that the high-dimensional features employed in deep learning refer to those with more than two dimensions. Specifically, the k-mer features utilized are three-dimensional, while the Position-Specific Scoring Matrix (PSSM) features are four-dimensional. In contrast, the features used in machine learning consist of two-dimensional matrices.”

Comments 3: Figure 2 does not specify which model (1 vs 2) represents deep learning or machine learning. Also, how can one tell that the differences between these two models are negligible or significant? Is there a way to calculate p values or any other alternatives based on the results obtained from repeated runs? It is especially important when the authors state that ‘the differences between the intersection and union of predictions from the two models are striking’, which, to me, are not visually ‘striking’. Figure 3 shares these issues as well.

Response 3: The figure captions have been revised to explicitly delineate the correspondence between the models and the results presented in the figures. While a single regression model can be evaluated using p-values, in the case of intersecting two models, the analysis base or sample set undergoes changes. As a result, quantile hit rate was employed for evaluation. Additionally, in practical promoter screening, results with intensity values in the higher quantiles are prioritized, making quantile hit rate a more suitable metric for promoter analysis and selection. The use of the term "astonishing" was indeed imprecise, and has been revised accordingly.

To explain this, we have modified Figure 2、Figure 3 and Section 2.2 by adjusting the sentences:“Figure 2. Analysis of Quantile Hit Rates (QHR) for the test set Using Deep Learning (model 1) and Machine Learning (model 2) Models.”、“Figure 3. Analysis of Quantile Hit Rates (QHR) for Out-of-Sample Mutant Promoters Using Deep Learning (model 1) and Machine Learning (model 2) Models.”、“In summary, notable differences are observed between the intersection and union of predictions generated by the two models.”
